# Left-handed metamaterial bandpass filter for GPS, Earth Exploration-Satellite and WiMAX frequency sensing applications

Md. Jubaer Alam[1]*, Eistiak Ahamed[2], Mohammad Rashed Iqbal Faruque[2], Mohammad Tariqul Islam[3], Ahmed Mahfuz Tamim[2]

**1** International University of Business Agriculture and Technology, Sector 10, Uttara Model Town, Dhaka, Bangladesh, **2** Space Science Centre (ANGKASA), Institute of Climate Change, Universiti Kebangsaan Malaysia, Bangi, Selangor, Malaysia, **3** Centre of Advanced Electronic and Communication Engineering, Universiti Kebangsaan Malaysia, Bangi, Selangor, Malaysia

* jubaer.alam@iubat.edu

**Data Availability Statement:** All relevant data are within the manuscript and its Supporting Information files.

## Abstract

Interferences and accuracy problem are one of the most talked issues in today's world for sensor technology. To deal with this contention, a microstrip framework consisting of a dual mode double negative (DNG) metamaterial based bandpass filter is presented in this article. To obtain the ultimate noise reduction bandpass filter, the proposed structure has to go through a series of development process, where the characteristics of the structure are tested to the limit. This filter is built on Rogers RT-5880 substrate with a 50Ω microstrip line. To pursue the elementary mode of resonant frequency, the ground layer of the structure is kept partially filled and a gradual analysis is executed on the prospective metamaterial (resonator) unit cell. Depending on the developed unit cell, the filter is constructed and fabricated to verify the concept, concentrating on GPS (1.55GHz), Earth Exploration-Satellite (2.70GHz) and WiMAX (3.60GHz) bands of frequencies. Moreover, the structure is investigated using Nicolson–Ross–Weir (NRW) approach to justify the metamaterial characteristics, and also tested on S-parameters, current distribution, electric and magnetic fields and quality factor. Having a propitious architecture and DNG characteristics, the proposed structure is suitable for bandpass filter for GPS, Earth Exploration-Satellite and WiMAX frequency sensing applications.

## Introduction

Nowadays web-accessible sensors become very popular in order to support the cyber physical system vision. This web-accessible sensor network and sensor data that can be discovered and access using the standard protocols and application program interface. However, this complicated web-accessible sensor technology has to deal with several interferences and data loss throughout the process. In this regard, the implementation of RF components and the use of metamaterials can have a significant impact on the sensor technology [1–2]. These elements mainly include power amplifiers [3] as active circuits and couplers [4], power dividers [5] and bandpass filters (BPFs) [6] as passive circuits. Designing BPFs is essential to meet the specific

**Funding:** This work was supported by the Research Universiti Grant, Universiti Kebangsaan Malaysia, Geran Universiti Penyelidikan (GUP), Code: 2018-134.

**Competing interests:** The authors have declared that no competing interests exist.

needs of industrial applications and consumer electronics. At present, numerous planar BPFs having lower insertion losses in the passbands and higher out-of-band rejections has been presented utilizing signal-interference techniques [7], stub loaded techniques, and so on. Moreover, balanced BPFs have more noise immunity advantages and low electromagnetic interference in wireless communication systems compared to their one-ended counterparts [8]. Now-a-days, limiting the parameters so that the operating frequency range works in a specific bandwidth has been a challenge. In this regard, one of the most deserving solutions to avoid interference is the metamaterial-based bandpass filter. The proper use of unorthodox electromagnetic properties of metamaterials can be one of the options to handle this complexity.

Metamaterial, a complex material synthetically engineered that exhibits some fascinating electromagnetic characteristics and overcomes the usual limitations of natural materials available. These extraordinary metamaterial properties may include negative permittivity, negative permeability, Snell's inverted law, as well as some similar reverse electromagnetic phenomena. With such electromagnetic phenomena, metamaterials can be utilized for applications including SAR reduction, super lenses, antenna design, filter [9–13], invisibility cloaking [14], electromagnetic absorbers [15], photovoltaic application [16] and electromagnetic bandgaps. A researcher named Wu et al. practised distinctive types of meta-surfaces at Terahertz (THz) frequencies, specifically. Some of them are tunable graphene [17], dual-band moiré meta-surface patches for multifunctional biomedical applications [18], and tunable multiband meta-surfaces [19]. Nevertheless, different structures and alphabet shaped metamaterials become very popular in different applications [20–22].

High-performance components and circuits are extensively desirable to build bandpass filters. The design of the BPFs is indispensable to meet the specific demands of consumer electronics and industrial applications. Recently, many planar BPFs with low insertion losses in the passbands and high out-of-band rejections have been reported using signal-interference techniques [7], stub loaded techniques, and so on. Additionally, balanced BPFs have more advantages of noise immunity and low electromagnetic interference in wireless communication systems compared with their single-ended counterparts [8]. In today's world, one of the most essential parameters are confinement among channels in a particular bandwidth. In spite of avoiding possible interferences in a system, passband filter is one of the competent arrangements to notify this problem by combining them to the system Jiang et al. 2017 [23]. Metamaterials can be a good option in this regards.

For the purpose of bandpass technology, Split ring resonators (SRR) or Double split ring resonators (DSRR) are commonly used. In this research, a multiple bandpass filter composed of split ring resonators with dual mode characteristics (dual mode resonators are used for double tuneable resonant circuit) is presented. To reduce the overall size of the filter, these dual mode resonators are used. A gradual step wise development is made to obtain the ultimate metamaterial resonator. The characteristics of the unit cell are verified on different aspects and the unit cell in implemented on a 1.575mm thick substrate Rogers RT-5880. Both sides of the structure are fed by metal strips (feedlines) and a correlation between input and output is made. A partial ground method is employed to support the structure and multiple parasitic (metal) strips are used to tune the circuit to its optimal value. It was a challenging task to use the resonators appropriately, so that the system maintains the balance between inductance and capacitance and operates at 1.55GHz, 2.70GHz and 3.60GHz respectively.

## Methodology

The main focus of this research is to construct a bandpass filter to reduce interferences from received signal. This filter structure has to be made in such a way so that it can be used in

transducers to pass the signal even further. In case of modulation (Amplitude and frequency), the basic signal needs to be carried by a carrier signal with higher frequencies. The overall modulated signal transmits from a transducer and receives by another transducer. The received signal in the transducer contains modulated signal with carrier as interferences. This received signal needs to be transmitted through the proposed filter structure, so that the interferences can be eliminated, and the output terminal gets the desired frequencies. The function of this bandpass filter would be to remove the interferences from the received signal and sense (allow) to pass signal at 1.55, 2.70 and 3.65GHz to the receiving terminal. To attain the ultimate bandpass filter, the overall system has to go through several stages. It starts with the development of a metamaterial unit cell and finishes with the proposed compact rectangular bandpass filter. Fig 1 exhibits the overall work flow of the entire system.

## Development of the metamaterial unit cell

Open loop resonators (OLRs) as metamaterials are studied in this research and their fundamental properties are investigated. A few perspectives regarding the coupling among OLRs are reviewed in this section. This OLR commonly known as microstrip OLR can be attained by folding a straight open resonator as shown in Fig 2(A). A qualitative analysis on this square open loop resonator can draw a comparison between full wave simulators and real distribution of electromagnetic fields.

The calculation of the resonant frequency was started from the input admittance at any point on a straight OLR within its length. Fig 2(B) shows the straight microstrip OLR and Fig 2(C) shows the equivalent circuit of it.

Admittance,

$$Y_{in} = jY_0(tan\theta_1 + tan\theta_2) = jY_o sin\theta_T cos\theta_1 cos\theta_2 \tag{1}$$

Where, output admittance is denoted by $Y_o$, and $\theta_T$ symbolizes total electrical length of the resonator, $\theta_T = \theta_1 + \theta_2$. If $Y_{in}$ becomes zero, it will create a standing wave. Therefore, there will be undetermined resonant frequencies at

$$\theta_T = n\pi \; or \; \mathfrak{l} = n\lambda/2 \tag{2}$$

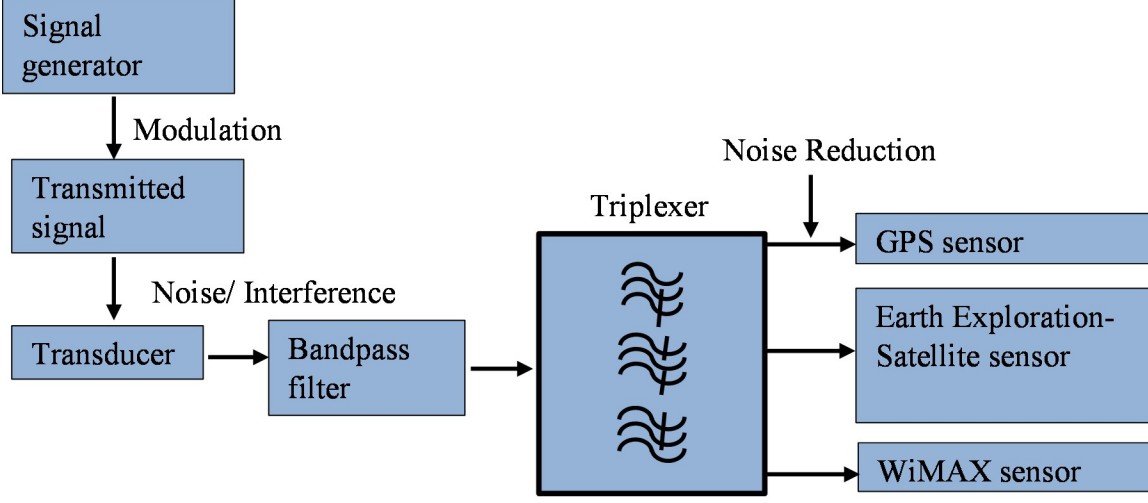

**Fig 1. Overall work flow of the entire system.**

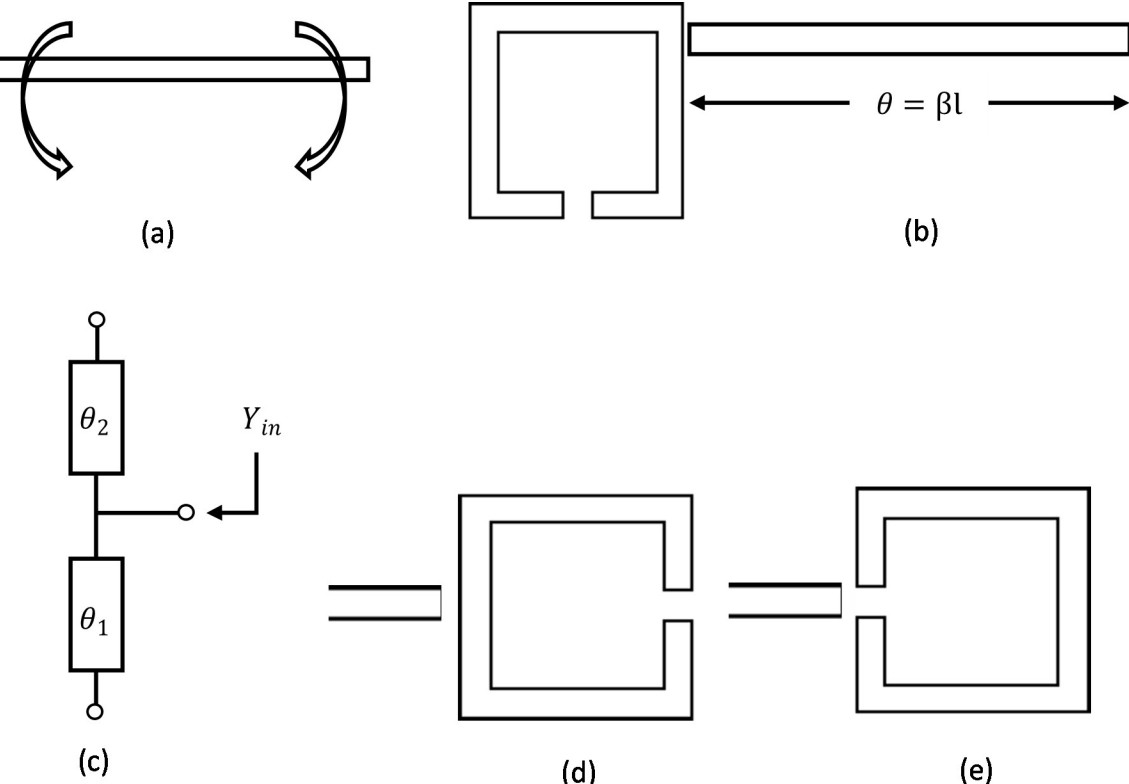

**Fig 2.** (a) Folding a straight OLR to the square OLR, Microstrip OLR (b) top view and (c) equivalent circuit, (d) Primary mode excitation and (e) both open ends excitation of the resonator.

At fundamental resonant frequency, $Y_{in}$ becomes infinity and voltage nulls can be found at, $\theta_1 = \theta_2 = \pi/2$. And in case of 2nd resonant frequency, voltage nulls are found when $\theta_1$ and $\theta_2$ become $\pi/2$ and $3\pi/2$, respectively. It is really important to know the voltage nulls as they decide the unexcited situations of the resonators. The point of feeding to the resonator is also important. It determines the mode of the resonator (even mode or odd mode). If the resonator is fed at the centre or via both free terminals, then the action of the resonator will be in even-mode operation. Fig 2D and 2E exhibit the excitation of the introduced resonator in an invalid of the fundamental mode and with respect to basic terminals, respectively.

The proposed metamaterial (open loop resonator) based filter design starts with a metamaterial unit cell. The unit cell structure is designed in the frequency range of 0.2-5GHz including its resonance. There are some notable strategies for metamaterial configuration to give synchronous negative permittivity and permeability utilizing SRRs [24]. The SRR is built utilizing two loops that are organized as two restricting concentric split rings. The SRR is a magnetically resonant structure that leads to a vertical magnetic field whose application produces negative permeability. The proposed design is modified by adding more resonators and splits. While designing this form of bandpass filter, both the single and dual-mode resonators are utilized, but the dual mode resonator is more suitable as it is a double-tunable resonant circuit [25]. The modification in the structure decreases the series capacitance loop resonator and increases the coupling between the inner and outer loops. The unit structure is printed over Rogers RT-5880 which thickness is about 1.575mm and 2.2 is the dielectric constant value. The main focus was to achieve a bandpass filter by combining a dual-mode OLR and a DS-SRR. After doing a series of evolution process the ultimate bandpass filter is achieved. Fig 3(A) shows the

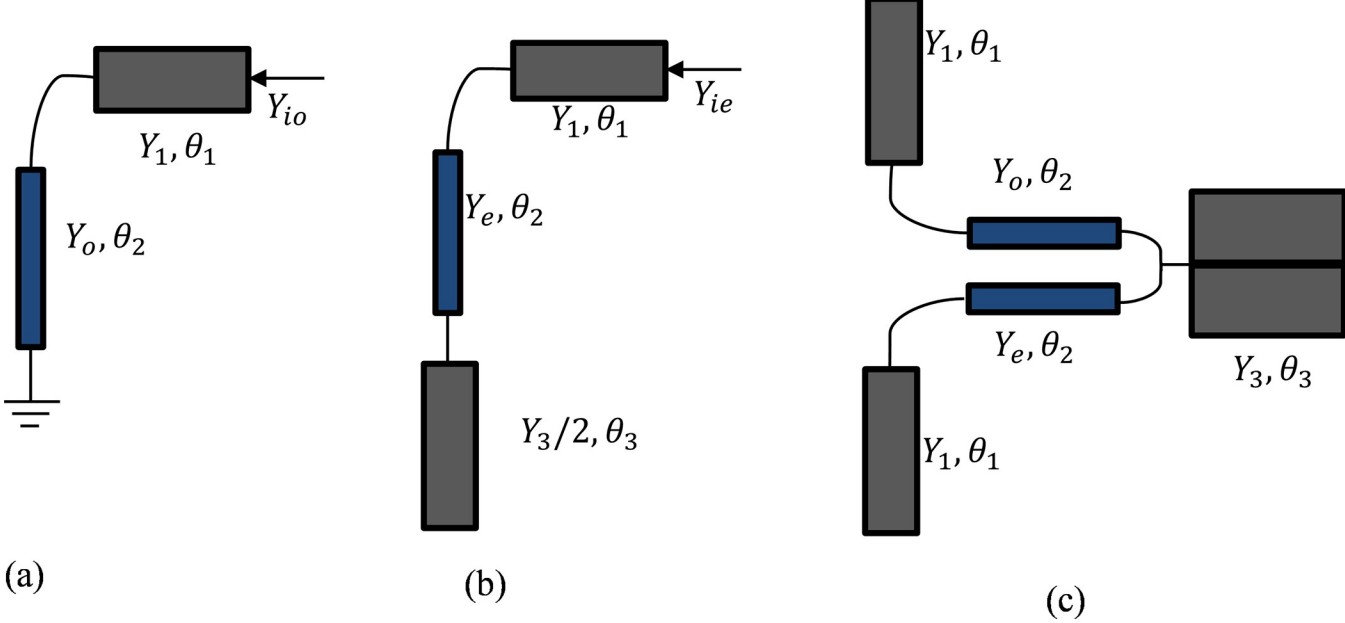

**Fig 3.** (a) Odd-mode equivalent circuit (b) even-mode equivalent circuit (c) dual-mode OLR.

equivalent circuit of the odd-mode, and Fig 3(B) shows the equivalent circuit of the even-mode. Combining this two gives the proposed dual-mode equivalent circuit, where the resonator works with in the frequency range, which is shown in Fig 3(C)

Resonant frequency can be determined by the input admittance. To operate in odd-mode excitation, the admittance ($Y_{io}$) is [25]:

$$Y_{io} = jY_1 \frac{Y_1 tan^2\theta - Y_o}{(Y_1 + Y_o)tan\theta} \tag{3}$$

At resonance, $Y_{io}$ becomes zero. Therefore,

$$tan^2\theta = K_1 = \frac{Y_o}{Y_1} \tag{4}$$

The fundamental frequency is $f_o$:

$$f_o = \frac{ctan^{-1}(\sqrt{K_1})}{2\pi L_1 \sqrt{\varepsilon_{eff}}} \tag{5}$$

Where, $c$ is the speed of light, $L_1$ is the microstrip line length and $\varepsilon_{eff}$ is the effective dielectric constant. To operate in even-mode excitation, the admittance ($Y_{ie}$) is:

$$Y_{ie} = jY_1 \frac{(2Y_1Y_e + Y_3Y_e + 2Y^2_e)tan\theta - Y_1Y_3tan^3\theta}{2Y_1Y_e - (Y_1Y_3 + Y_3Y_e + 2Y^2_e)tan^2\theta} \tag{6}$$

At resonance,

$$tan^2\theta = K_2 = \frac{Y_e}{Y_1} + \frac{2Y^2_e}{Y_1Y_3} + \frac{2Y_e}{Y_3} \tag{7}$$

The fundamental frequency is $f_e$:

$$f_e = \frac{ctan^{-1}(\sqrt{K_2})}{2\pi L_1 \sqrt{\varepsilon_{eff}}} \tag{8}$$

Therefore, by knowing $K_1$, $K_2$, and θ, it is obvious that resonance frequency can be achieved.

## S-parameters for unit cell

Fig 4 exhibits the simulation set up and the gradual evaluation of the proposed resonator. The resonator is tested on the electromagnetic field to verify its working range of frequency. The finite integration technique-based CST Microwave Studio simulation software was used for the numerical examination of the resonator. The two wave-guide ports were put at the positive and negative end, on the z-axis of the resonator, and the electromagnetic wave was spread between the ports. Moreover, the PEMC (Perfect Electric-Magnetic Conductor) boundary condition was adopted for simulation. For simulation, frequency domain solver was used. Moreover, for the analysis purpose of this synopsis, a tetrahedral mesh is being utilized inclusion of a flexible mesh.

The ultimate unit cell is obtained by an extensive eight step parametric study, where the cell shows a concrete transmission pole at 3.11GHz. Fig 4(A) shows the simulation set up for the unit cell and Fig 4(B) shows the step wise evaluation process. The results of step 7 and 8 are quite similar, but more attenuation is obtained in step 8, which makes it the desired unite cell for the proposed bandpass filter.

## Metamaterial characteristics analysis of the resonator

There are plenty of methods to characterize metamaterial, like Nicolson-Ross-Weir (NRW) method [26], Transmission-reflection (TR) method [27] etc. In this study, NRW method was adopted for metamaterial characterization. The simplified equations of NRW method is as follows-

$$V_1 = S_{21} + S_{11} \tag{9}$$

$$V_2 = S_{21} - S_{11} \tag{10}$$

$$\mu_r \approx \left(\frac{2}{jk_0 d}\right)\left(\frac{1 - V_2}{1 + V_2}\right) \tag{11}$$

$$\varepsilon_r \approx \left(\frac{2}{jk_0 d}\right)\left(\frac{1 - V_1}{1 + V_1}\right) \tag{12}$$

$$\eta = \sqrt{(\varepsilon_r \mu_r)} \tag{13}$$

where, $j$ is the imaginary operator, $\varepsilon_r$ is the effective permittivity, $\mu_r$ is the effective permeability, 'd' is the thickness of the substrate, $k_0$ is the wave number and η is the refractive index.

Fig 5 shows the effective parameters of the unit cell on 0 to 360-degree azimuthal angle rotation with 60-degree interval. Fig 5(A) exhibits 0 and 360 degree effective parameters for rotating position. Throughout this process, the unit cell is tested on relative permittivity, permeability and refractive index. It is clear from Fig 5 that the unit cell exhibits negative characteristics on all the three parameters and possesses double negative behaviour at the

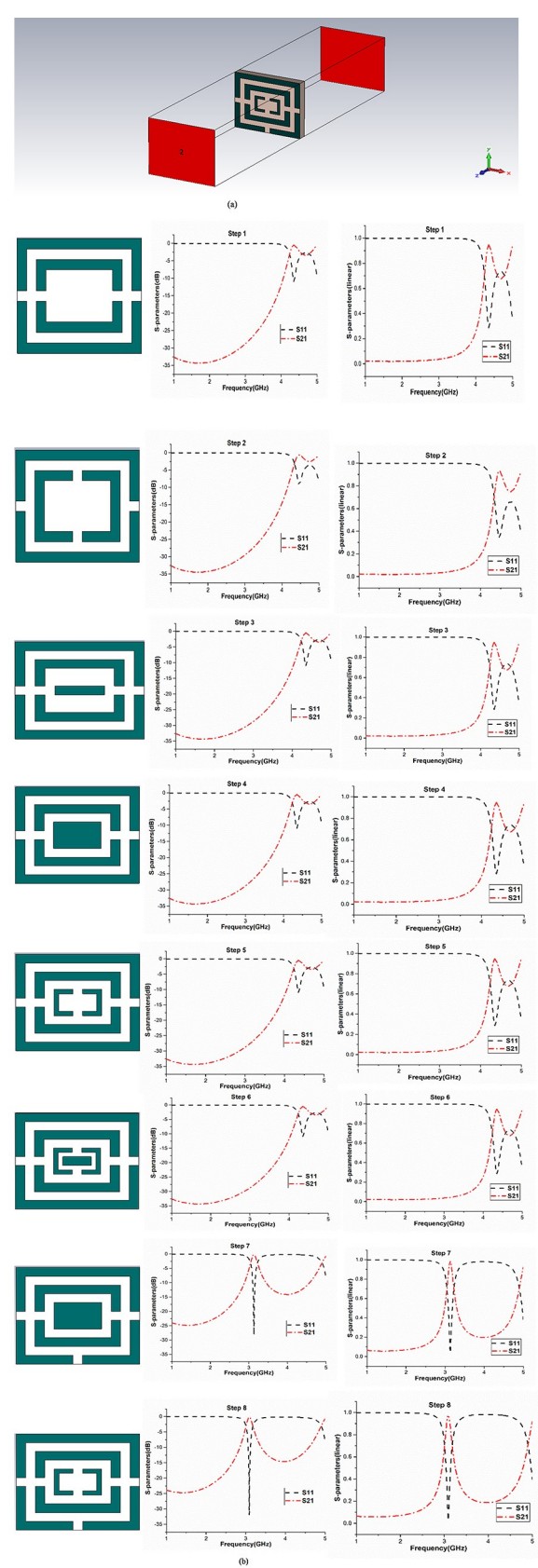

**Fig 4.** (a) Simulation set up for the metamaterial unit cell, (b) Step wise scattering parameters analysis of the resonator where left column represents logarithmic scale and right column represents linear scale.

resonating frequency. Table 1 shows the performance analysis of the unit cell at different azimuthal rotation angles.

At 3.11GHZ (where the unit cell shows bandpass characteristic), the values of the effective parameters are negative for all the azimuthal angles of the unit cell. Thus, the metamaterial unit cell can be called as the Double negative metamaterial at 3.11GHz.

## Construction of bandpass filter

The fundamental resonator interaction procedure is attention-grabbing as a result of its particular coupling framework and minimal structure for filter applications. The principal goal of this introduced triple split ring resonator (TSRR) is to achieve passband frequencies of 1.55GHz, 2.70GHz and 3.60GHz. To achieve this goal, an incomplete ground is connected on the opposite part of the filter; two compound TSRRs are put in a steady progression including 180˚ spin. A flat sum delay is created between non-neighbouring components in cross-coupled filters for open-circle resonators. Both TSRRs are associated with two microstrip feedlines. Four parasitic metal lines are stretched onto the dielectric material to increase the electrical length despite rising length outcomes in resonance to lower frequencies where the current is distributed throughout the patch. Nevertheless, with the increase in frequency, the current needs to cover curvier ways, in contrast to the straight one at lower frequencies in view of the skin effect due to which current drifts through a conductor's surface instead of its essential part at upper frequencies [27]. These parasitic streaks of metal cause the filter to operate at lower frequencies. Agilent N5227A vector network analyser was used to determine the filter parameters, and the Agilent N4694-60001 electronic calibration module was used for adjustment. Fig 6(A) and 6(B) demonstrate the geometrical perspectives front and back of the filter, gradually.

Fig 7 represents the fabricated prototype, and Table 2 records the parameters of the filter. Thickness of the OLR substrate is 1.575mm and its dielectric constant value is 2.20. In Table 2, all the Ws (W-W10) are the width wise measurements of the filter, Ls (L-L6) are the measurements of length-by-length, gs (g-g2) represents the gaps, and an indicates supporting lines width covering OLRs.

Fig 8(A) represents the S-parameters except parasitic strips, where all the frequencies of resonance are beyond the center frequencies (1.65, 2.58, and 4.09GHz, respectively). In Fig 8(B), using the parasitic strips, all the frequencies of resonance are initiated at 1.55, 2.70, 3.60GHz, respectively. For the filter, to verify the insertion and return losses, different resonators parameters were studied.

Fig 8(C) represents the individual characteristics of the insertion and return losses for both the measured and simulated results. The fraction detachment in the measured values might be occurred owing to the connector among the feed and the controller. However, the measured results followed the simulated results. The effect of this detachment is very little.

Fig 9 depicts the surface current distribution of the proposed structure (without and with parasitic strips). The metal strips or parasitic strips used in the structure work as the resonator at the optimum frequencies. The purpose of using these metal strips is to allow the bandpass filter to operate in lower frequencies. Fig 9 exhibits the effect of metal strips on the filter structure by the current distribution. When a time changing magnetic field penetrates through the metallic rings, it induces an EMF by Faraday's Law of electromagnetic induction which in turn produces a rotation current. The ring produces its own magnetic field which enhances the incident field. Hence there is an inductive effect associated with the ring.

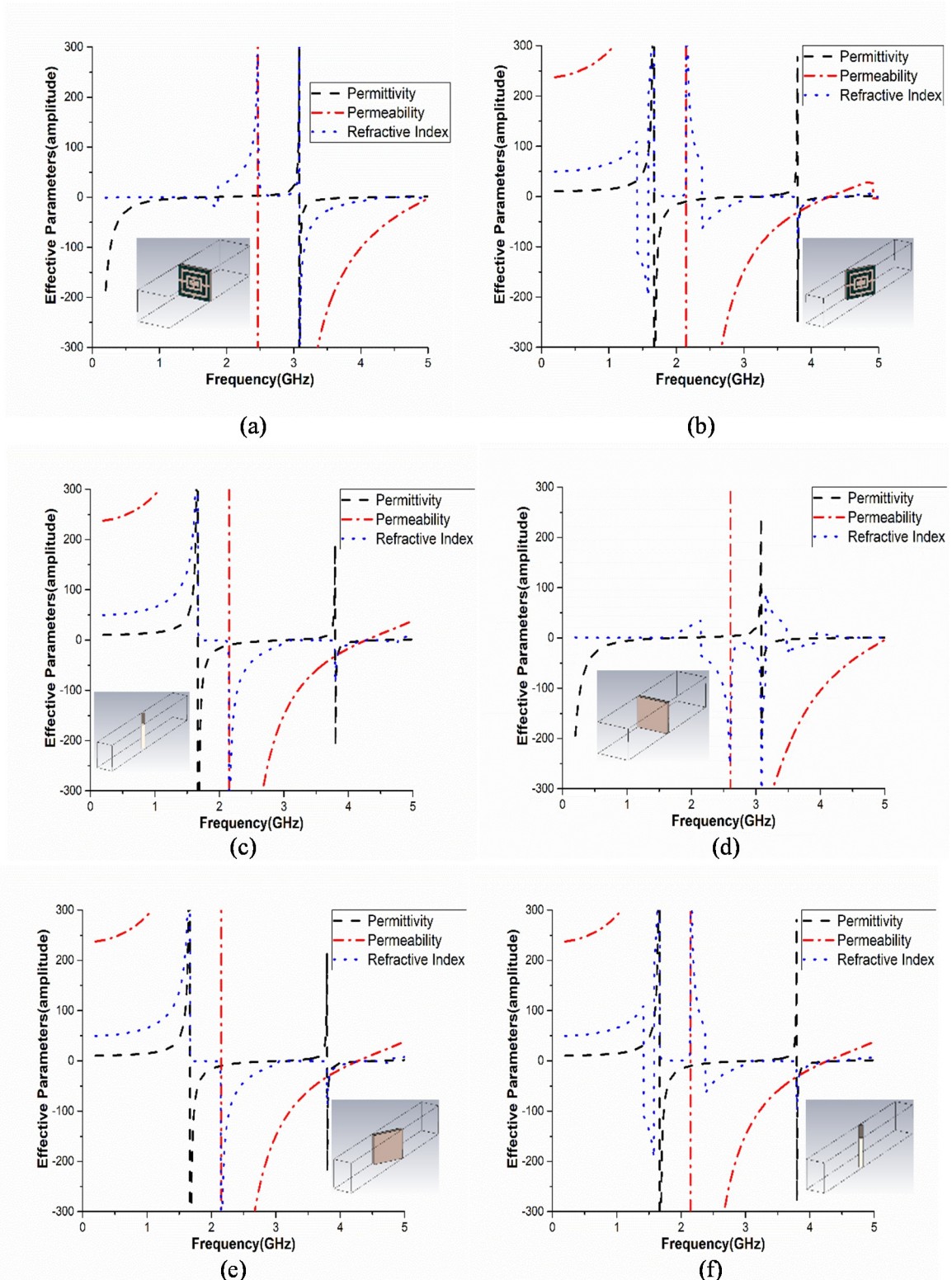

**Fig 5.** Amplitude of the effective parameters at (a) 0˚ and 360˚, (b) 60˚, (c) 120˚, (d) 180˚ (e) 240˚ and (f) 300˚ azimuthal rotation angles.

**Table 1. Performance analysis of the unit cell at different azimuthal rotation angles.**

| Angle (φ) | Permittivity (ε) | Permeability (μ) | Refractive Index (η) | Amplitude at 3.11GHz | | | BW of DNZ (GHz) |
|---|---|---|---|---|---|---|---|
| | | | | ε | μ | η | |
| **0° = 360°** | 0.2–1.69, 3.08–4.25 | 2.46–5.0 | 0.2–1.87, 3.09–5.0 | -57.95 | -406.46 | -158.66 | 1.17 |
| **60°** | 1.66–3.25, 3.80–4.55 | 2.15–4.28 | 1.42–1.58, 2.38–4.28 | -0.40 | -119.28 | -0.08 | 0.87 & 0.47 |
| **120°** | 1.67–3.18, 3.8–4.54 | 2.15–4.28 | 1.69–4.77 | -0.94 | -120.99 | -0.06 | 1.03 & 0.48 |
| **180°** | 0.2–1.75, 3.08–4.35 | 2.61–5.0 | 2.15–3.14, 3.47–3.93 | -45.99 | -349.52 | -129.92 | 0.07 & 0.46 |
| **240°** | 1.66–3.17, 3.80–4.54 | 2.15–4.28 | 1.69–4.77 | -0.38 | -119.95 | -0.06 | 1.02 & 0.48 |
| **300°** | 1.66–3.16, 3.80–4.55 | 2.14–4.28 | 1.42–1.58, 2.38–4.28 | -0.39 | -119.28 | -0.08 | 0.78 & 0.48 |

The capacitance part comes because of the splits in the ring which effectively behaves as a parallel plate capacitor. The capacitance created because of the splits prevents the flow of current over the rings; the capacitance between rings facilitates current flow around the structure. A number of such split ring resonators are arranged in a periodic array such that EM waves interact with them as a homogenous medium. By modifying the arrangement of the rings, a medium having an effective negative permeability for adjustable range of frequencies can be obtained. In addition, the electric and magnetic resonances overlap to one another creates effective negative permittivity. Therefore, the effective negative refractive index is created when the overlapping occurred. Fig 9 shows the flow of similar pattern current through inner and outer rings exhibiting the bandpass behavior at the resonating frequencies.

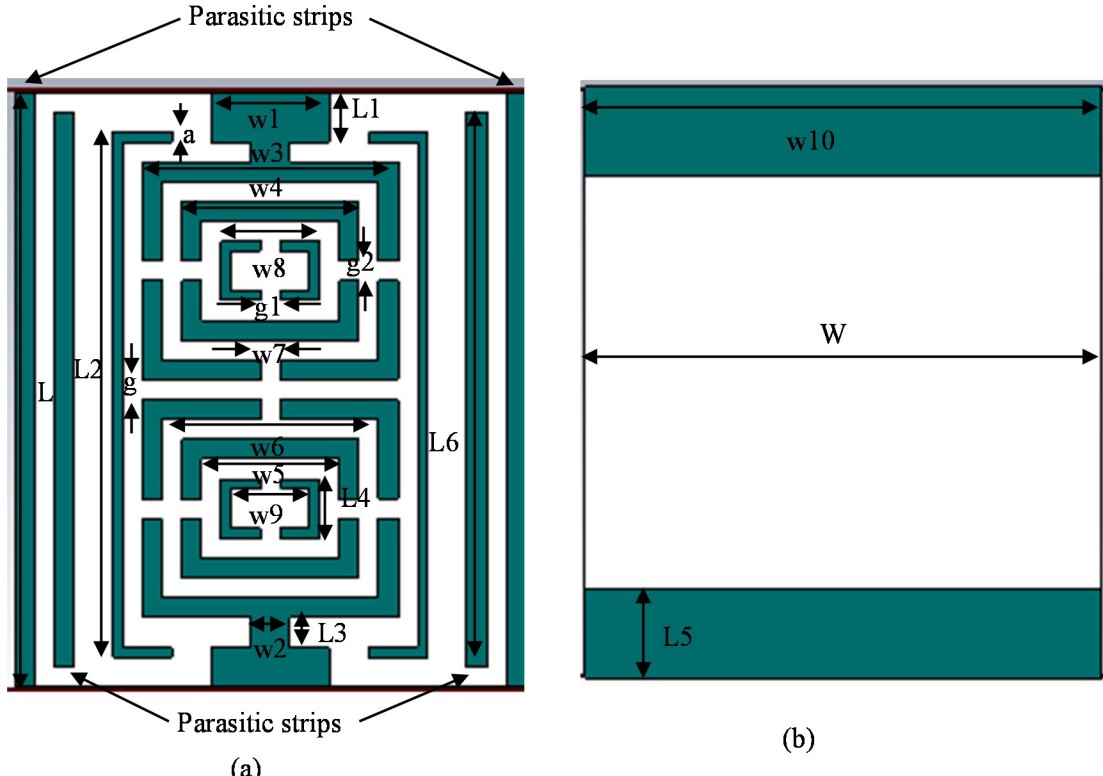

**Fig 6.** Geometrical arrangements of the introduced passband filter (a) Front view and (b) Back view.

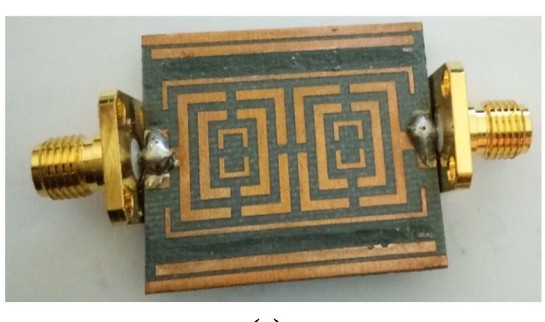
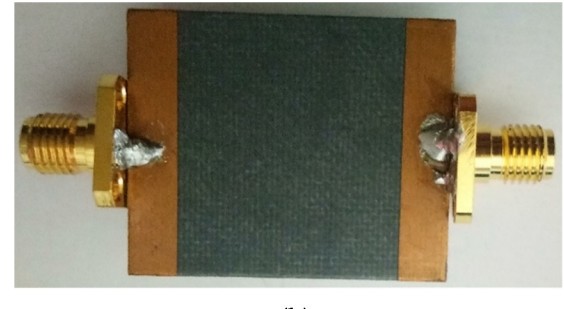

(a)                                                                                          (b)

**Fig 7.** Fabricated prototype (a) Front and (b) Back views

The effective parameters analysis shows that they are also at the frequencies of resonance. Negative effective permeability of the structural formation is affected by the series of gaps. Adding those gaps originates a narrow band gap incessantly; as a result, permittivity and permeability emerged close to the frequencies of resonance [14].

$$\varepsilon_r = \frac{c}{j\pi f d} \times \frac{(1 - V_1)}{(1 + V_1)} \tag{14}$$

$$\mu_r = \frac{c}{j\pi f d} \times \frac{(1 - V_2)}{(1 + V_2)} \tag{15}$$

$$\eta_r = \frac{c}{j\pi f d} \times \sqrt{\frac{(S_{21} - 1)^2 - S_{11}^2}{(S_{21} + 1)^2 - S_{11}^2}} \tag{16}$$

Fig 10 represents the real and imaginary values of the effective permittivity, permeability, and refractive index, gradually. It is clear that the values of real and imaginary of the effective parameters are at the center frequencies (1.55, 2.70, and 3.60GHz) of the passbands.

Group delay is defined as the rate of change of transmission phase angle with respect to frequency, which is shown in Fig 11. In filter, group delay is very important to show the bandpass accuracy i.e. signal is passing without any dispersion or loss. The group delay of a filter is nearly proportional to its order. On the other hand, filter group delay is inversely proportional to filter bandwidth i.e. small percentage bandwidth filters have large group delay. In our developed filter, we found resonance frequency at 1.55, 2.70 and 3.60 GHz respectively, whereas we found group delay pick at our desired frequency in simulation, which proves that the

**Table 2. Introduced passband filter parameters.**

| Parameters | Dimensions (mm) | Parameters | Dimensions (mm) |
|---|---|---|---|
| a | 0.5 | w5 | 7 |
| W, w10 | 26 | w6 | 11 |
| L | 30 | w7, g, g1, g2 | 1 |
| w1 | 6 | w8 | 5 |
| w2, L3 | 2 | w9 | 4 |
| w3 | 13 | L1, L4 | 3 |
| w4 | 9 | L2 | 25 |
| L5 | 4.5 | L6 | 28 |

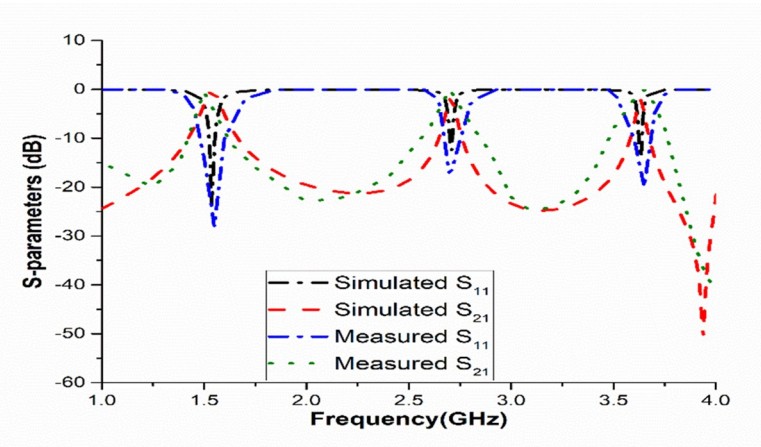

**Fig 8.** Scattering parameters of the filter (a) without and (b) with parasitic strips (c) Measured and simulated S-parameters.

performance of bandpass filter is enough good that it can pass the desired frequency with less noise and less loss. However, measured results followed the simulated group delay shown despite of some minor detachment. This discrepancy due to some possible reasons like there may be a calibration error associated with the Agilent N5227 vector network analyzer, impedance mismatch between the SMA connectors and the feedline, minor fabrication error, some insertion loss due to minor soldering problems of SMA connectors or there might be some reflective metallic components for what some reflection exits in the measurement which caused the variation of the results etc.

## Analysis for sensor application

The proposed structure is further tested on quality factor or Q-factor [28]. This Q-factor defines the filter characteristics of any RLC circuit. It is the reciprocal of power factor. It can be formulated as the ratio between the resonant frequency and the bandwidth of the RF resonant circuit.

$Q = F_0/F$

Q-factor for 1.55GHz = (fc/BW) = 1.55/ (1.59–1.48) = 14.09

Q-factor for 2.70GHz = (fc/BW) = 2.70/ (2.75–2.66) = 30

Q-factor for 3.60GHz = (fc/BW) = 3.60/ (3.66–3.56) = 36

At all the three resonant frequencies, the values of Q-factor are 14 or more indicate the bandpass characteristics of the structure. Fig 12(A) shows the quality factor of the structure. Bandpass characteristics of a resonator can be analyzed on insertion loss and return loss. Insertion loss is the attenuation of a passive device, which can be calculated by the ratio of output power to input power in logarithmic scale. The main concern of this research is to find out the higher sensitivity with lower bandwidth. The structure shows high values of quality factor with minimum bandwidth.

Therefore, the insertion loss = 10log (Pout/Pin)

At 1.55GHz, the insertion loss is -0.025dB.

Pout = Pin– 0.025 ≈ Pin; 99.97% signal can pass through the filter and the system loss is minimum. In case of 2.70 and 3.60GHz, the insertion losses are -0.92dB and -1.5dB respectively. That turns to be more than 97% propagation or less than 3% system loss. Besides, the return loss plays a vital role as well. It happened because of the impedance mismatch. It can be

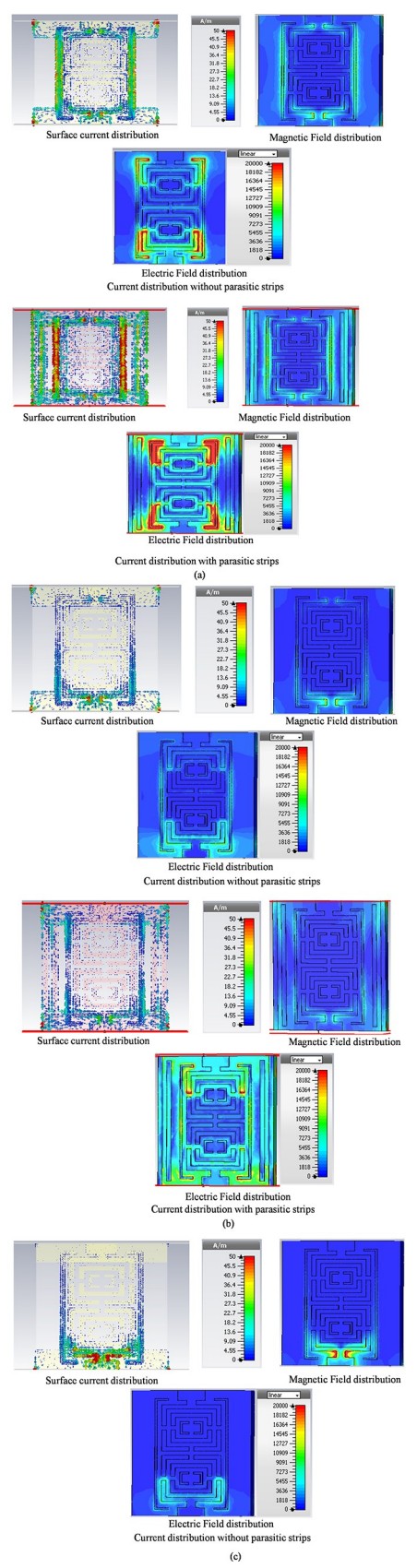

Surface current distribution    Magnetic Field distribution

Electric Field distribution
Current distribution without parasitic strips

Surface current distribution    Magnetic Field distribution

Electric Field distribution
Current distribution with parasitic strips
(a)

Surface current distribution    Magnetic Field distribution

Electric Field distribution
Current distribution without parasitic strips

Surface current distribution    Magnetic Field distribution

Electric Field distribution
Current distribution with parasitic strips
(b)

Surface current distribution    Magnetic Field distribution

Electric Field distribution
Current distribution without parasitic strips
(c)

**Fig 9.** Current distribution without and with parasitic strips at (a) 1.55GHz (b) 2.70GHz and (c) 3.65GHz.

defined as the reflected power towards the source because of the mismatch. It can be calculated by the ratio between reflected powers to input power of the system in logarithmic scale.

Therefore, the return loss = 10log (Pr/Pin)

At 1.55GHz, the return loss is -23.6dB. Pr = Pin– 23.6≈ (Pin/$2^8$) ≈0.0039Pin; the reflected power is very negligible, and the amount is 0.39% of the input power. In case of 2.70 and 3.60GHz, the return losses are -12.5dB and -13.93dB respectively. That turns to be more than 95.6% propagation or less than 4.4% system loss. In addition, to justify the sensitive nature of the structure, the response of the equivalent circuit of the filter is tested in ADS. A signal generator is placed at one end and a receiver at the other, the receiver shows the entire transmitted signal without any interruption from 0 to 4GHz (Fig 12B). Then the equivalent circuit of the structure is placed in between the transmitter and the receiver (Fig 12C), and the monitor shows only 1.56, 2.70 and 3.60GHz at receiving end justifies the application of the sensor at GPS, Earth Exploration-Satellite and WiMAX frequency bands.

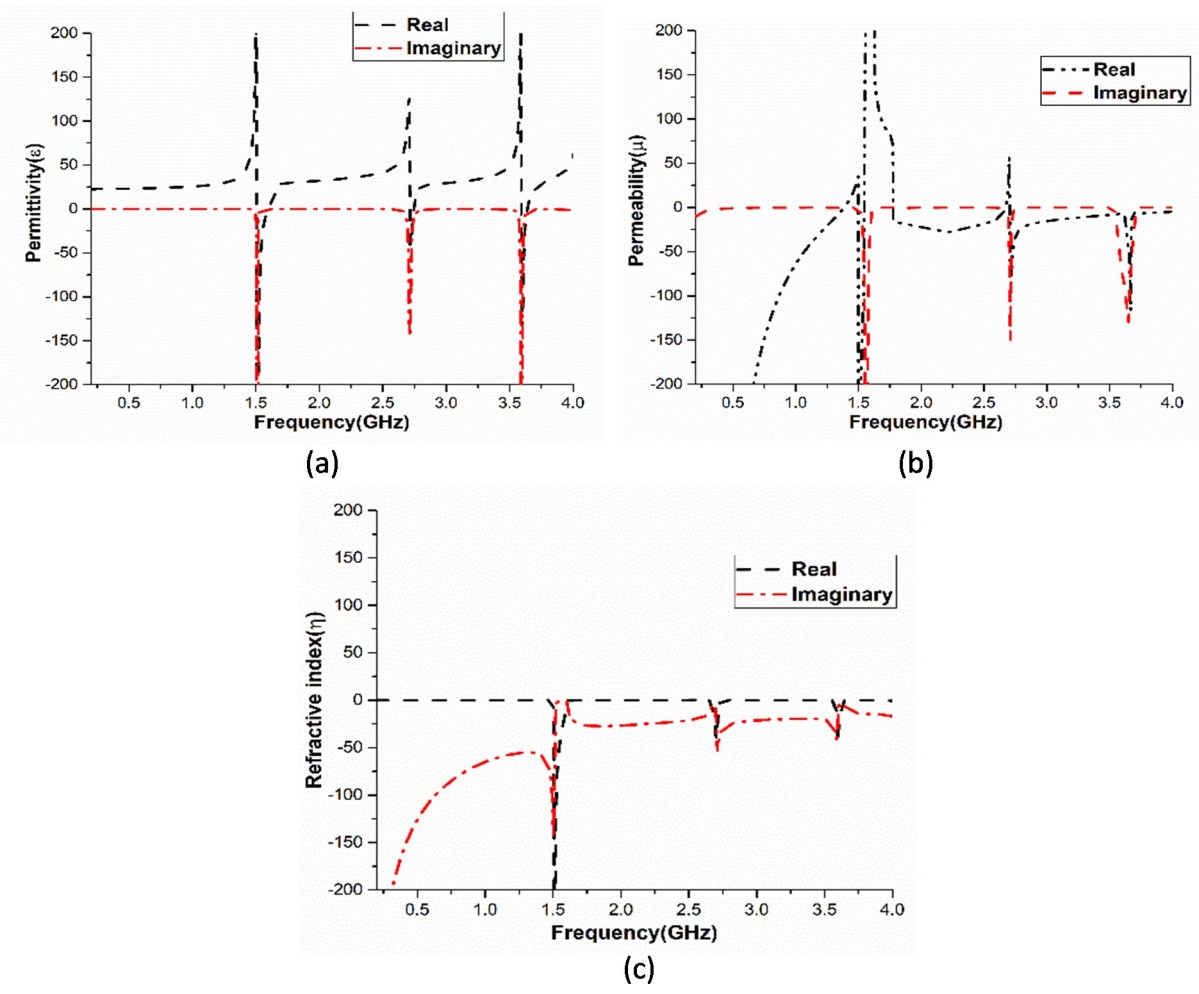

**Fig 10.** Effective parameters (a) Effective permittivity, (b) Effective permeability and (c) Effective refractive index.

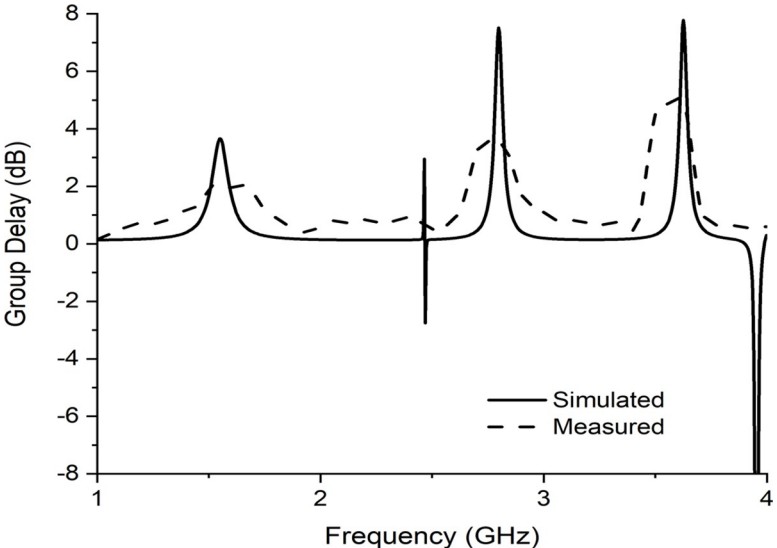

**Fig 11. Simulated and measured group delay for the passband filter.**

From the results of ADS, it is evident that, before using the proposed structure, the transducer uses to receive all the signals at its range, but when the equivalent circuit is introduced, it removes the interferences and the transmitting signal restricts to 1.56, 2.70 and 3.60GHz and more likely to behave like sensors. The performance comparison of proposed bandpass filter with existing works is shown in Table 3.

Xu et al. (2018) proposed a dual mode resonator based high-selectivity bandpass filter, where the center frequency of the filter is 2.1 GHz. The filter operates in S-band but the length of structure is 79 mm. Gorur et al. (2014) also presented a metamaterial based 27 mm bandpass filter with center frequencies at 2.45 and 3.5 GHz respectively. Though the filter is suitable for dual band operation, but still it does not have dual mode and does not support GPS or IOT application. Athukoral et al. (2018) and Hua et al. (2011) worked on dual mode resonators separately. Both the filters operate in L and S-band frequencies, but the size of the filters are higher than the proposed filters (more than 30 mm). However, the filters are applicable for controllable transmission zeros and tunable transmission notch respectively. Hong et al. (2007) proposed a dual mode resonator based four pole bandpass filter with operating frequencies 0.5 to 2.5 GHz. The filter operates well for dual mode microstrip filter but it cannot be used multi band applications. All the bandpass filters mentioned above are highly effective in their particular applications, but neither of them cover more than two bands like the proposed filter does. Moreover, the proposed filter in the combination of metamaterial based dual mode resonators with double negative characteristics at the filtering frequencies, which is absent in the above mentioned works. Hence, the smaller size of the filter and metamaterial characteristics made it unique from the other bandpass filters.

## Conclusion

In this research, a dual mode new metamaterial resonator is designed by the step by step development process in the expected 1 to 5GHz frequency range and the resonator is tested for metamaterial characteristics as well as bandpass features. A microstrip structure on perfect dielectric substrate for bandpass filter is developed, and the novel metamaterial is embedded on the microstrip structure for bandpass applications with compact size, with better

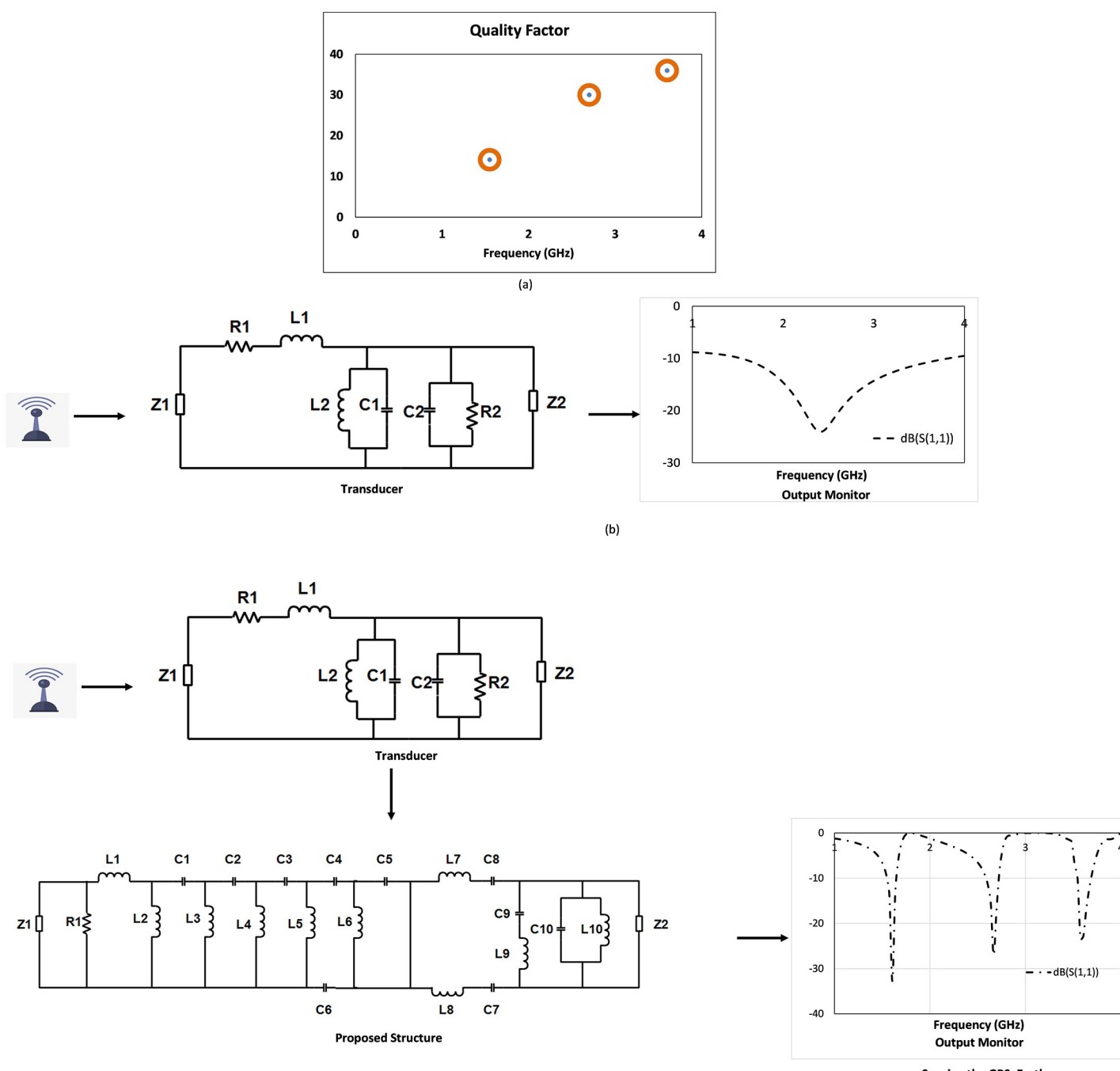

**Fig 12.** (a) Quality factor vs. Frequency, schematic diagram of the microwave sensor (b) without using the proposed structure and (c) after using the proposed structure.

performance and double negative characteristics. Sufficient number of parasitic strips is applied to the microstrip structures according to their applicable resonant frequency (1.55, 2.70 and 3.60GHz). The structure is tested on scattering parameters, effective parameters, quality factor, and noise reduction sensing ability, where the measured result of the fabricated prototype follows the simulated one. The filter proposed in this article exhibits bandpass

**Table 3. Comparison of the proposed bandpass filters with previous works.**

| Previous works | Speciallity | Size (mm) | Opt./Cent. Freq.(GHz) | Application |
|---|---|---|---|---|
| Xu et al. 2018 | Dual mode resonators with multiple Stubs | 79 | 2.1 | High-selectivity bandpass filters |
| Gorur et al. 2014 | Metamaterial based filter | 27 | 2.45, 3.5 | WLAN and Wimax |
| Athukorala et al. 2018 | Dual mode open loop microstrip resonators | 40 | 0–4 | Controllable transmission zeros |
| Hong et al. 2007 | Four pole with dual mode resonators | 34 | 0.5–2.5 | Dual mode microstrip filter |
| Hua et al. 2011 | Dual mode stub-loaded resonator | 54 | 0–4.5 | Tunable notch |
| Proposed Filters | Triple split ring resonators | 30 | 1.5, 2.70, 3.60 | GPS, IOT, WLAN and WiMAX |

characterized with double negative characteristics and good time domain behaviours which make it very suitable for being used in practical GPS, Earth Exploration-Satellite and WiMAX frequency sensing applications.

## Supporting information

**S1 File. Design methodology.**
(PDF)

## Acknowledgments

This work was supported by the Research Universiti Grant, Universiti Kebangsaan Malaysia, Geran Universiti Penyelidikan (GUP), Code: 2018–134.

## Author Contributions

**Conceptualization:** Md. Jubaer Alam.

**Formal analysis:** Md. Jubaer Alam, Eistiak Ahamed.

**Funding acquisition:** Mohammad Rashed Iqbal Faruque.

**Methodology:** Md. Jubaer Alam.

**Software:** Md. Jubaer Alam, Eistiak Ahamed.

**Supervision:** Mohammad Rashed Iqbal Faruque.

**Validation:** Mohammad Tariqul Islam, Ahmed Mahfuz Tamim.

**Visualization:** Md. Jubaer Alam.

**Writing – original draft:** Md. Jubaer Alam.

**Writing – review & editing:** Eistiak Ahamed, Mohammad Rashed Iqbal Faruque.

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
