## [Decision Letter · Decision Letter 0]

16 Oct 2019

Left-Handed Metamaterial Bandpass Filter for GPS, Earth Exploration-Satellite and WiMAX Frequency Sensing Applications

PONE-D-19-26520

Dear Dr. Alam,

We are pleased to inform you that your manuscript has been judged scientifically suitable for publication and will be formally accepted for publication once it complies with all outstanding technical requirements.

With kind regards,

Mohammad Maktoomi, PhD

Academic Editor

PLOS ONE

Additional Editor Comments (optional):

Please take care of the following and reviewers' comments before the final submission of the file.

(1) Please re-read the entire manuscript and make corrections to grammatical errors/spelling mistakes. For example, in the Abstract section, 2nd line, please use "consisting" instead of "consists of".

The first sentence below Fig. 1 in the subsection " Development of the Metamaterial Unit Cell": please provide the full-form of OLR as Open-loop resonator (OLR). 

(2) In general, each variable must be defined before they are used in an equation or immediately following the equation. For example, Y0 must be defined in equation 1, etc.

(3) what is Agilent N4694-60001? A calibration kit? If yes, please mention that.(4) Please redraw some of the figures as the text is blurred/not clear on many of them: For example, in Fig. 12, you can use a free program called xcircuit to draw nice-looking schematics. Elements names, such as L1, C1 etc should be clear, there is no need to draw a plot using the dotted line as there is just one plot in that figure. No need to show the legend for the same reason. It is better to import the ADS/CST plot data to MATLAB or any other such program to plot the graphs professionally.

Reviewers' comments:

Reviewer's Responses to Questions

**Comments to the Author**

1. Is the manuscript technically sound, and do the data support the conclusions?

Reviewer #1: Yes

Reviewer #2: Yes

2. Has the statistical analysis been performed appropriately and rigorously? 

Reviewer #1: Yes

Reviewer #2: Yes

3. Have the authors made all data underlying the findings in their manuscript fully available?

Reviewer #1: Yes

Reviewer #2: Yes

4. Is the manuscript presented in an intelligible fashion and written in standard English?

Reviewer #1: Yes

Reviewer #2: Yes

5. Review Comments to the Author

Reviewer #1: The authors have addressed all of my queries. In addition, the authors have done good job incorporating the suggestions in the revised manuscript which has improved the quality of the manuscript. But authors should mention OLR full form and even-mode resonance condition of Yie = 0 before going to equation (7) in the manuscript. After it, I think it is ready for publication.

Reviewer #2: The authors have addressed all my concerns. Also, the current manuscript version has been revised very well according to all the reviewers' comments. So I think it can be as accepted for the Journal PlosOne.

6. PLOS authors have the option to publish the peer review history of their article (what does this mean?). If published, this will include your full peer review and any attached files.

Reviewer #1: No

Reviewer #2: Yes: Kai-Da Xu

---

## [Editor Report · Acceptance letter]

1 Nov 2019

PONE-D-19-26520 

Left-Handed Metamaterial Bandpass Filter for GPS, Earth Exploration-Satellite and WiMAX Frequency Sensing Applications 

Dear Dr. Alam:

I am pleased to inform you that your manuscript has been deemed suitable for publication in PLOS ONE. Congratulations! Your manuscript is now with our production department. 

With kind regards,

on behalf of

Dr. Mohammad Maktoomi 

Academic Editor

PLOS ONE